# Incidence of extraovarian clear cell cancers in women with surgically diagnosed endometriosis: A cohort study

Liisu Saavalainen[1], Heini Lassus[1], Anna But[2], Mika Gissler[3,4], Oskari Heikinheimo[1]*, Eero Pukkala[5,6]

**1** Department of Obstetrics and Gynaecology, University of Helsinki and Helsinki University Hospital, Helsinki, Finland, **2** Department of Public Health, Biostatistics Consulting, University of Helsinki and Helsinki University Hospital, Helsinki, Finland, **3** Information Services Department, Finnish Institute for Health and Welfare (THL), Helsinki, Finland, **4** Department of Neurobiology, Care Sciences and Society, Karolinska Institutet, Stockholm, Sweden, **5** Finnish Cancer Registry–Institute for Statistical and Epidemiological Cancer Research, Helsinki, Finland, **6** Faculty of Social Sciences, Tampere University, Tampere, Finland

* oskari.heikinheimo@helsinki.fi

## Abstract

### Background

Endometriosis is associated with increased risk of clear cell ovarian cancer and has even suggested being an etiological factor for this cancer. Association between endometriosis and extraovarian clear cell cancers is unclear. This study aimed to assess the association between surgically diagnosed endometriosis and risk of extraovarian clear cell cancers according to the type of endometriosis (i.e., ovarian, peritoneal, and other endometriosis) and the site of clear cell cancer.

### Methods

In this register-based historic cohort study we identified all women with surgically diagnosed endometriosis from the Finnish Hospital Discharge Registry 1987–2012. Data on extraovarian clear cell cancers of these women were obtained from the Finnish Cancer Registry. The follow-up started January 1st, 2007 or at endometriosis diagnosis (if later), and ended at emigration, death or on the December 31st, 2014. Standardized incidence ratios were calculated for each site of clear cell carcinoma (intestine, kidney, urinary tract, gynecological organs other than ovary), using the Finnish female population as reference.

### Results

The endometriosis cohort consisted of 48,996 women, including 22,745 women with ovarian and 19,809 women with peritoneal endometriosis. Altogether 23 extraovarian clear cell cancers were observed during 367,386 person-years of follow-up. The risk of extraovarian clear cell cancer was not increased among all women with surgically diagnosed endometriosis (standardized incidence ratio 0.89, 95% confidence interval 0.56–1.33) nor in different types of endometriosis. The incidence of clear cell cancer in any specific site was not increased either.

**Data Availability Statement:** The Finnish register data have been given for this specific study, and the data cannot be shared without authorization from the register keepers. More information on the

authorization application to researchers who meet the criteria for access to confidential data can be found at Findata, the Health and Social Data Permit Authority (www.findata.fi).

**Funding:** The Hospital District of Helsinki and Uusimaa has funded this research project. The first author (LS) has received salary for a week to be able to write this article. The funder had no role in study design, data collection and analysis, decision to publish, or preparation of the manuscript.

**Competing interests:** The authors have declared that no competing interests exist.

## Conclusions

The risk of extraovarian clear cell cancers in women with surgically diagnosed endometriosis is similar to that in the general population in Finland.

## Introduction

The association between endometriosis and ovarian cancer has been intensively studied. Endometriosis has especially been associated with ovarian cancer of endometrioid and clear cell histology in cohort studies [1, 2]. Some studies have described a spatial and chronological association with transformation of atypical endometriosis to ovarian cancer and, molecular studies have detected similar genetic alterations (e.g. mutations of ARID1A and PIK3CA) in endometriosis and in ovarian cancer of clear cell histology [3]. Thus, accumulating data suggest that endometriosis is an etiological factor for clear cell cancer.

In addition to ovaries, cancer of clear cell histology is also seen in other sites such as urinary tract, kidney, uterus, colorectum and peritoneal cavity [4–9]. Thus, the possible association between endometriosis and extraovarian clear cell cancer is of interest. First, endometriosis is causing systemic inflammation, and may be situated anywhere in peritoneum or located deep in other tissues in- and outside of abdominal cavity, e.g. in bowel, bladder and pleura [10, 11]. Second, similar molecular changes have been described in cancers and in endometriosis situated also outside the ovary [6, 12–14]. Third, the urogenital system is embryologically and anatomically intimately interwoven.

The aim of this study was to assess the risk of extraovarian clear cell cancer in women with surgically diagnosed endometriosis in a large population-based cohort in Finland. In addition, we examined this possible risk according to the site of clear cell cancer and type of endometriosis (i.e., ovarian, peritoneal and other endometriosis).

## Materials and methods

The ethics committee of the Hospital District of Helsinki and Uusimaa (238/13/03/03/2013) approved the study. The data were fully anonymized. As this is a register-based study, no informed consent was required. In addition, we obtained permissions from the Finnish Institute for Health and Welfare (THL/546/5.05.00/2014), the Digital and Population Data Services Agency (D1794/410/14), and Statistics Finland (TK53-547-14) for the data and linkages. There was no public or patient involvement of study design or interpretation of results. We obtained funding for the study from the research funds of the Hospital District of Helsinki and Uusimaa.

The cohort formation and the quality assessment of the surgically diagnosed endometriosis identified from the Finnish Hospital Discharge Register from 1987 to 2012 have been described previously [15]. All first diagnoses of endometriosis (International Statistical Classification of Diseases and Related Health Problems [ICD], 9th Revision [1987–1995]: 6171A, 6172A, 6173A, 6173B, 6174A, 6175A, 6176A, 6178X, 6179X; 10th Revision [1996–2012]: N80.1-N80.6, N80.8, N80.80, N80.81, N80.89, N80.9) except adenomyosis set in a procedure, either as main or subsidiary diagnosis, were included. The endometriosis cohort consisted of 48,996 women. The division into the types of endometriosis, ovarian (n = 22,745), peritoneal (n = 19,809) and other endometriosis (n = 6,442) was performed by using the diagnostic codes assigned at the index procedure (S1 Table).

The Finnish Cancer Registry maintains a database of all diagnosed cancers, including dates of cancer diagnoses, and topography and morphology codes of all malignancies in Finland from 1953 forward. The cancer data are collected by health care professionals, pathology laboratories, hospitals and institutions providing cancer treatment from all over the country [16]. Data on the histologically verified extraovarian clear cell carcinomas in the cohort of surgically diagnosed endometriosis were assessed by linkage to the Finnish Cancer Registry using the personal identity codes. All topography codes except ovary (C56), morphology codes for clear cell adenocarcinoma (8310) and clear cell adenocarcinofibroma (8313), and behavior code 3 for malignant neoplasms according to the coding by International Classification for Diseases in Oncology, 3rd edition were used when retrieving the data.

The coding by Classification for Diseases in Oncology, 3rd edition at the Finnish Cancer Registry began in 2007, which made it possible to distinct the incident extraovarian clear cell cancers. We therefore started follow-up from the January 1st, 2007, or from the endometriosis diagnosis, which ever was later. The follow-up ended on the day of emigration, death, or on December 31st, 2014 which ever came first. The dates on emigrations and deaths were retrieved from the Finnish Population Register Centre. All linkages were done by using women's unique personal identity codes, which are available for all Finnish citizens and permanent residents since the 1960s.

Person-years of follow-up were calculated by five-year age categories and calendar periods, and by time since the index day (<5, and ≥5 years). The standardized incidence ratio (SIR) was calculated for each site of clear cell carcinoma (i.e., intestine, kidney, urinary tract and gynecological organs other than ovary) as the ratio between the observed and the expected number of cancers in each stratum. The expected number was defined by multiplying the accumulated person-years of follow-up in each stratum by the corresponding cancer incidence rate in the entire Finnish female population. These reference rates for each five-year age category and five-year calendar period were calculated from the data base of the Finnish Cancer Registry. The 95% confidence intervals for the standardized incidence ratio were based on the assumption that the number of observed cases followed a Poisson distribution.

## Results

The numbers of women by age groups in the endometriosis cohort and according to the types of endometriosis at the time of surgically diagnosed endometriosis are presented in Table 1. In addition, Table 1 shows the numbers of person-years by age at follow-up. There were altogether 367,386 person-years of follow-up of which 52.1% were in ages 50 years or higher. The mean follow-up time was 7.5 years.

**Table 1. Extraovarian clear cell cancers among women with surgically diagnosed endometriosis: Numbers of women (n) at the time of diagnosis and person-years (py) by age (years) at follow-up.**

| | TYPE OF ENDOMETRIOSIS | | | | | | | |
| | All | | Ovarian | | Peritoneal | | Other | |
| Age | n (%) | py (%) | n (%) | py (%) | n (%) | py (%) | n (%) | py (%) |
|---|---|---|---|---|---|---|---|---|
| 10–29 | 4,669 (9.5) | 14,426 (3.9) | 1,846 (8.1) | 5,400 (3.2) | 1,881 (9.5) | 6,196 (4.1) | 942 (14.6) | 2,830 (6.1) |
| 30–39 | 11,352 (23.2) | 64,806 (17.6) | 4,824 (21.2) | 27,177 (16.0) | 4,417 (22.3) | 24,953 (16.5) | 2,111 (32.8) | 12,677 (27.2) |
| 40–49 | 13,490 (27.5) | 96,607 (26.3) | 6,069 (26.7) | 41,572 (24.5) | 5,755 (29.1) | 41,547 (27.5) | 1,666 (25.9) | 13,489 (28.9) |
| 50–59 | 13,123 (26.8) | 101,112 (27.5) | 6,546 (28.8) | 47,847 (28.2) | 5,379 (27.2) | 43,159 (28.6) | 1,198 (18.6) | 10,106 (21.7) |
| 60–69 | 5,829 (11.9) | 76,072 (20.7) | 3,140 (13.8) | 39,789 (23.4) | 2,212 (11.2) | 29,909 (19.8) | 477 (7.4) | 6,374 (13.7) |
| ≥70 | 533 (1.1) | 14,362 (3.9) | 320 (1.4) | 8,108 (4.8) | 165 (0.8) | 5,081 (3.4) | 48 (0.8) | 1,174 (2.5) |
| Total | 48,996 (100.0) | 367,386 (100.0) | 22,745 (100.0) | 169,892 (100.0) | 19,809 (100.0) | 150,843 (100.0) | 6,442 (100.0) | 46,650 (100.0) |

**Table 2. Extraovarian clear cell cancers among women with surgically diagnosed endometriosis according to the type of endometriosis by age (years), follow-up time (years) and cancer site.** Observed number of cancer cases (O), standardized incidence ratio (SIR) and 95% confidence interval (CI).

| | | TYPE OF ENDOMETRIOSIS | | | | | | | |
| --- | --- | --- | --- | --- | --- | --- | --- | --- | --- |
| | | All | | Ovarian | | Peritoneal | | Other | |
| | | (n = 48,996) | | (n = 22,745) | | (n = 19,809) | | (n = 6,442) | |
| | | O | SIR (95%CI) | O | SIR (95%CI) | O | SIR (95%CI) | O | SIR (95%CI) |
| **Age** | | | | | | | | | |
| | 10–29 | 0 | 0.00 (0.00–170) | 0 | 0.00 (0.00–455) | 0 | 0.00 (0.00–399) | 0 | 0.00 (0.00–868) |
| | 30–39 | 0 | 0.00 (0.00–21.4) | 0 | 0.00 (0.00–50.7) | 0 | 0.00 (0.00–56.8) | 0 | 0.00 (0.00–107) |
| | 40–49 | 2 | 0.71 (0.09–2.56) | 2 | 1.64 (0.20–5.93) | 0 | 0.00 (0.00–3.02) | 0 | 0.00 (0.00–9.77) |
| | 50–59 | 7 | 0.96 (0.39–1.98) | 2 | 0.58 (0.07–2.09) | 4 | 1.29 (0.35–3.30) | 1 | 1.38 (0.04–7.71) |
| | 60–69 | 11 | 0.92 (0.46–1.65) | 7 | 1.12 (0.45–2.31) | 4 | 0.86 (0.23–2.19) | 0 | 0.00 (0.00–3.70) |
| | ≥70 | 3 | 0.81 (0.17–2.36) | 0 | 0.00 (0.12–1.75) | 1 | 0.76 (0.02–4.25) | 2 | 6.45 (0.78–23.3) |
| **Follow-up time** | | | | | | | | | |
| | <5 | 0 | 0.00 (0.00–4.79) | 0 | 0.00 (0.00–8.02) | 0 | 0.00 (0.00–20.5) | 0 | 0.00 (0.00–28.4) |
| | ≥5 | 23 | 0.92 (0.58–1.37) | 11 | 0.87 (0.44–1.56) | 9 | 0.88 (0.40–1.67) | 3 | 1.30 (0.27–3.81) |
| **Cancer site (ICD-10)[a]** | | | | | | | | | |
| | Intestine (C17-21, C26)[b] | 0 | 0.00 (0.00–114) | 0 | 0.00 (0.00–229) | 0 | 0.00 (0.00–277) | 0 | 0.00 (0.00–1260) |
| | Cervix (C53) | 0 | 0.00 (0.00–9.27) | 0 | 0.00 (0.00–18.8) | 0 | 0.00 (0.00–22.8) | 0 | 0.00 (0.00–93.0) |
| | Uterus (C54) | 1 | 0.61 (0.02–3.40) | 0 | 0.00 (0.00–4.37) | 1 | 1.56 (0.04–8.69) | 0 | 0.00 (0.00–24.1) |
| | Kidney (C64) | 19 | 0.82 (0.49–1.27) | 9 | 0.77 (0.35–1.45) | 7 | 0.75 (0.30–1.54) | 3 | 1.37 (0.28–4.00) |
| | Urinary organs (C65-68) | 0 | 0.00 (0.00–77.7) | 0 | 0.00 (0.00–149) | 0 | 0.00 (0.00–197) | 0 | 0.00 (0.00–901) |
| **All** | | 23 | 0.89 (0.56–1.33) | 11 | 0.84 (0.42–1.50) | 9 | 0.87 (0.40–1.64) | 3 | 1.23 (0.25–3.59) |

ICD-10 –International Statistical Classification of Diseases and Related Health Problems version 10.

[a]Three cases with ill-defined sites were not classified here: two were coded as gynaecologic cancers but documented as kidney cancers, one case had no topography.

[b]Includes cancers located in small intestine, colon, rectum, rectosigmoid, rectum, anus, digestive organs other than oesophagus, stomach, liver, gallbladder, pancreas.

Altogether 23 extraovarian clear cancers were observed in women with history of surgically diagnosed endometriosis; the expected number of cases based on the incidence rate in comparable women of the Finnish female population was 26. The incidence of extraovarian clear cell cancer in women with endometriosis was not increased in any age category, specific type of endometriosis, or specific site of cancer (intestine, cervix, uterus, kidney, urinary organs). All cancer diagnoses had been made more than 5 years after the date of endometriosis surgery (Table 2).

## Discussion

The overall incidence of extraovarian clear cell cancer in Finnish women with surgically diagnosed endometriosis did not differ from the incidence of same-aged women in the entire female population in Finland, nor did the incidence of clear cell cancer in any specific site of cancer. The incidence was not increased in any specific type of endometriosis either.

We found extraovarian cancers of clear cell histology to be very rare after surgically diagnosed endometriosis in Finland, only 23 extraovarian cancers were identified in the follow-up of 370,000 person-years. The two previous studies focusing on clear cell cancers in extraovarian sites with concurrent diagnosis of endometriosis of the same location have been based on pathological review of cases in single institutions [17, 18]. In these studies clear cell and endometrioid carcinomas as well as adenosarcoma have been the most common types of cancer types diagnosed concurrently with extraovarian endometriosis. More recently, a survey-based cohort study from Japan–where the proportion of clear cell ovarian cancer is exceptionally high–found one clear cell cancer case among 1397 women with concurrent endometriosis outside the ovaries and the pelvic peritoneum [19]. Our study is the first to evaluate the standardized incidence ratios of extraovarian clear cell cancer on population-based registries after surgically diagnosed endometriosis, hence the direct comparison of our findings with the earlier studies is not possible.

Clear cell histology accounts for 90% of all kidney cancers [20]. A Swedish population-based study found a 1.4-fold increased incidence of kidney cancer in women with endometriosis while no increased risk of kidney cancer was found in our earlier study [21, 22]. Clear cell histology is seen in only 1–5% of endometrial carcinomas and in ≤1% of all cervical cancers [7]. Clear cell cancers of kidney, uterine corpus and cervix share similar molecular changes with endometriosis [6, 7]. Therefore, a common etiology between these cancers has been suggested [23]. Clear cell cancer of cervix and vagina has been associated with the prenatal exposure to diethylstilbestrol in the 1950s and 1960s [24]. However, the use of diethylstilbestrol was minimal in Finland [25]. In the present study, we found no increase in the incidence of clear cell cancer in kidneys, corpus or cervix of the uterus.

In line with our results, in the recent studies the incidence of the rare clear cell cancer of the bladder or urethra was not increased in women with endometriosis [4, 5]. Colorectal cancer of clear cell histology is a rarity in women [26]. A recent literature review concluded that the colorectal clear cell cancer in women with endometriosis has specific immunohistochemical characteristics and behavior when compared to the other primary clear cell colorectal cancers [8]. Common pathogenesis of residual embryological Müllerian nests was suggested. Nevertheless, the incidence of intestinal clear cell cancer was not increased in our study. There are case reports of women with endometriosis who concurrently or later had been diagnosed with clear cell cancer in various locations–such as in vagina, rectovaginal septum, peritoneum, diaphragm, and scars such as abdominal wall, umbilical node, caesarean section, or in episiotomy, or in unknown primary sites [9, 27–34].

A strength of our study is the large, population-based cohort, which comprises almost 370,000 person-years of follow-up. The Finnish Hospital Discharge Register and Finnish Cancer Registry are known for their completeness and high quality [16, 35]. In a bidirectional quality assessment of the study cohort 179 of 184 cases (97%) the endometriosis diagnosis verified from the patient files had been correctly reported to the Finnish Hospital Discharge Register. In addition, among the 168 patients who were operated for deep infiltrating endometriosis, 159 (95%) were found among the cohort identified from the Finnish Hospital Discharge Register [15].

Restricting the study to surgically diagnosed endometriosis allowed us to assess the risks according to the type of endometriosis, especially ovarian and peritoneal endometriosis. Our results can be generalized only to women with surgically diagnosed endometriosis. The histological diagnosis of clear cell cancer can be difficult, which may lead to the diagnosis of unknown or undifferentiated histology thus causing underestimation of true number of cases in our study. However, since the observed and expected numbers of cancers are derived from the same database, possible shortcomings in completeness of coding should not markedly affect our main result estimates, SIRs.

The coding of histology according to precise ICD-O-3 nomenclature started at Finnish Cancer Registry in 2007, which enabled the use of clear cell cancer histological data only after 2007. Thus, out of the large cohort, only women whose endometriosis diagnosis was set in 2007–2012 contribute person-years to follow-up category of less than five years after surgical diagnosis of endometriosis. Due to the diagnosis of endometriosis commonly in fertile age, the number of follow-up years is also small in the oldest age-groups. In these strata the statistical power to detect an association is low. Also, we did not have the data of the possible hormonal or non-steroidal anti-inflammatory medications often used in the treatment of endometriosis, which might have had an effect to the results. In the study by Modesitt et al. patients with extraovarian cancers arising in endometriosis were more likely to be postmenopausal and use hormone replacement therapy [18].

We might lack some cases in which extraovarian clear cell cancer and endometriosis were diagnosed at the same procedure. In clinical work, diagnosis of cancer may lead to undercoding of coexistent endometriosis in surgery as the malignant disease is the main diagnosis. If such cases of endometriosis diagnoses lack from our cohort, it would lead to lower SIRs in follow-up years under 5 years of surgical endometriosis diagnosis but would not affect the SIR in follow-up after 5 years of surgical endometriosis diagnosis.

## Conclusions

We found no increased risk of extraovarian clear cell cancer in women with surgically diagnosed endometriosis. Surgically diagnosed endometriosis seems to be a risk factor only for clear cell cancer of the ovary.

## Supporting information

**S1 Table. Type of endometriosis according to the International Statistical Classification of Diseases and Related Health Problems versions 9 (ICD-9) and 10 (ICD-10).**
(DOCX)

## Author Contributions

**Conceptualization:** Liisu Saavalainen, Heini Lassus, Oskari Heikinheimo, Eero Pukkala.

**Data curation:** Liisu Saavalainen, Anna But, Mika Gissler, Eero Pukkala.

**Formal analysis:** Eero Pukkala.

**Supervision:** Oskari Heikinheimo, Eero Pukkala.

**Writing – original draft:** Liisu Saavalainen.

**Writing – review & editing:** Heini Lassus, Anna But, Mika Gissler, Oskari Heikinheimo, Eero Pukkala.

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
