## [Decision Letter · Decision Letter 0]

31 Mar 2021

PONE-D-21-03577

Incidence of extraovarian clear cell cancers in women with surgically diagnosed endometriosis: a cohort study

PLOS ONE

Dear Dr. Heikinheimo,

Thank you for submitting your manuscript to PLOS ONE. After careful consideration, we feel that it has merit but does not fully meet PLOS ONE’s publication criteria as it currently stands. Therefore, we invite you to submit a revised version of the manuscript that addresses the points raised during the review process.

Three Reviewers have assessed this manuscript. Their observations on the scientific reliability of the study are very different. The Authors should carefully consider the issues raised and revise the manuscript accordingly.

We look forward to receiving your revised manuscript.

Kind regards,

Paola Viganò

Academic Editor

PLOS ONE

Journal Requirements:

2. In your ethics statement in the manuscript and in the online submission form, please ensure that you have discussed whether all data/samples were fully anonymized before you accessed them and/or whether the IRB or ethics committee waived the requirement for informed consent. If patients provided informed written consent to have data/samples from their medical records used in research, please include this information.

Reviewers' comments:

Reviewer's Responses to Questions

**Comments to the Author**

1. Is the manuscript technically sound, and do the data support the conclusions?

Reviewer #1: Yes

Reviewer #2: No

Reviewer #3: Partly

2. Has the statistical analysis been performed appropriately and rigorously? 

Reviewer #1: Yes

Reviewer #2: I Don't Know

Reviewer #3: Yes

3. Have the authors made all data underlying the findings in their manuscript fully available?

Reviewer #1: Yes

Reviewer #2: Yes

Reviewer #3: No

4. Is the manuscript presented in an intelligible fashion and written in standard English?

Reviewer #1: Yes

Reviewer #2: No

Reviewer #3: Yes

5. Review Comments to the Author

Reviewer #1: This research article is very interesting and answers one important question in endometriosis and oncology.

The document is well written with a clear aim and methodology.

The research includes a large group population and this obviously contributes on the strength of the research

All the current researches on this topic have been correctly cited in the article and discussed on the discussion session.

Reviewer #2: The present study aims at evaluating the association between extraovarian clear cell carcinomas and the diagnosis of endometriosis. I have several concerns regarding the consistency of this study and in my opinion the results and conclusions do not reliably represent an original finding on this topic.

- Some of the limitations of the present study are typical of all studies derived from population based national registries

o the diagnosis/coding might not be punctual and patient follow up might not be precise

o most commonly only the main diagnosis is indicated (as stated by the Authors themselves)

o Histopathological review in all these cases is not available. This procedure in case of clear cell histotype, as for other rare tumors, should be mandatory in order to provide a reliable analysis.

- Another concern is that the reported cases of clear cell tumors might not be really associated with endometriosis (the definition of endometriosis associated tumors in not simply the coexistence of these two conditions but has to fulfill precise histopathological criteria).

- I do not understand the relevance and usefulness of Table 1 for data and results interpretation.To me data analysis , result interpretation , as present in this paper, is not very eady for the reader.

Reviewer #3: I read manuscript entilted "Incidence of extraovarian clear cell cancers in women with surgically diagnosed endometriosis: a cohort study" with great interest. Endometriosis is known risk factor for ovarian clear cell cancer but there is no easy pathological explanation for possible influence of endometriosis on extraovarian clear cell cancer and why it even should be. And it was also proven by the Authors. But this main conculusion should be also supported by data from "general population".

Please describe general population - how it was choosen? what was age structure of general population? what was the incidence of extraovarian clear cell cancer in general population?

I found only statement that "The risk of extraovarian clear cell cancers in women with surgically diagnosed endometriosis is similar to that in the general population." but I did not find any data from general population or clarifications in disussion which can prove this thesis.

The other main point to correct is that this study refers only to finnish population. Please state this clearly in abstract and in the main text. The results of similar study conducted in Asiatic populations may have different results, because of the known higher incidence of clear cell cancer among Asians.

The explanation in discussion section about Finnish Hospital Discharge Register and Finnish Cancer Registry I found very important and supportive.

6. PLOS authors have the option to publish the peer review history of their article (what does this mean?). If published, this will include your full peer review and any attached files.

Reviewer #1: No

Reviewer #2: No

Reviewer #3: **Yes: **Maria Szubert, MD

---

## [Author Response · Author response to Decision Letter 0]

9 May 2021

Thank you for the opprotunity to revise our manuscript. 

-We have made some stylistic changes in the manuscript according to the journal requirements. 

-The ethics committee of Hospital District of Helsinki and Uusimaa (238/13/03/03/2013) approved the study. The data were fully anonymized. As this is a registry based study, no informed consent was required.

-The Hospital District of Helsinki and Uusimaa has funded this research project. The first author (LS) has received salary for a week to be able to write this article. The funder had no role in study design, data collection and analysis, decision to publish, or preparation of the manuscript.

-The data of the study are available upon request. There are legal restrictions on sharing this de-identified data set, imposed by Finnish Institute for Health and Welfare. The possible request can be made in www.findata.fi.

- And thank you for your comments on our manuscript. Please find enclosed our revised manuscript entitled “The incidence of non-ovarian clear cell cancer in women with surgically diagnosed endometriosis: a cohort study”, a manuscript with tract changes, and our point-by-point responses to reviewers’ comments. We tried to do our best to answer and renew the manuscript.

---

## [Editor Report · Decision Letter 1]

19 May 2021

PONE-D-21-03577R1

Incidence of extraovarian clear cell cancers in women with surgically diagnosed endometriosis: a cohort study

PLOS ONE

Dear Dr. Heikinheimo,

Thank you for submitting your manuscript to PLOS ONE. After careful consideration, we feel that it has merit but does not fully meet PLOS ONE’s publication criteria as it currently stands. Therefore, we invite you to submit a revised version of the manuscript that addresses the points raised during the review process.

ACADEMIC EDITOR: The manuscript has been improved. There is a minor issue that should be addressed. The Authors extensively discussed the types of cancer detected in endometriosis patients in the Discussion but, apart from the table, the Results section does not mention this topic at all. A discussion should be referred to the findings observed. A brief description of the cancer types observed would be useful also in the Results section.

Abstract: there is a repetition in the Results

We look forward to receiving your revised manuscript.

Kind regards,

Paola Viganò

Academic Editor

PLOS ONE
---

## [Author Response · Author response to Decision Letter 1]

25 May 2021

Thank you for your comments concerning our manuscript.

Please find enclosed our replies and revisions:

ACADEMIC EDITOR: 

1) The manuscript has been improved. There is a minor issue that should be addressed. 

The Authors extensively discussed the types of cancer detected in endometriosis patients in the Discussion but, apart from the table, the Results section does not mention this topic at all. A discussion should be referred to the findings observed. A brief description of the cancer types observed would be useful also in the Results section.

1) Thank you. Results section very briefly summarizes all our results shown in Table 2. 

To highlight that we also studied the specific sites of clear cell cancer added some words in the text. (page 8, lines 156-157)

"Altogether 23 extraovarian clear cancers were observed in women with history of surgically diagnosed endometriosis; the expected number of cases based on the incidence rate in comparable women of the Finnish female population was 26. The incidence of extraovarian clear cell cancer in women with endometriosis was not increased in any age category, specific type of endometriosis, or specific site of cancer (intestine, cervix, uterus, kidney, urinary organs). All cancer diagnoses had been made more than 5 years after the date of endometriosis surgery. (Table 2)" 

In Discussion the first section presented other similar-like studies, the second association with clear cell cancers in kidney, uterine and cervix cancer, and the third bladder, urethra and intestinal cancers. 

We would like to keep that in our manuscript.

ACADEMIC EDITOR:

2) Abstract: there is a repetition in the Results

2) We have corrected this repetition.

---

## [Editor Report · Decision Letter 2]

2 Jun 2021

Incidence of extraovarian clear cell cancers in women with surgically diagnosed endometriosis: a cohort study

PONE-D-21-03577R2

Dear Dr. Heikinheimo,

We’re pleased to inform you that your manuscript has been judged scientifically suitable for publication and will be formally accepted for publication once it meets all outstanding technical requirements.

Kind regards,

Paola Viganò

Academic Editor

PLOS ONE

Additional Editor Comments (optional):

The manuscript is now suitable for publication
---

## [Editor Report · Acceptance letter]

18 Jun 2021

PONE-D-21-03577R2 

Incidence of extraovarian clear cell cancers in women with surgically diagnosed endometriosis: A cohort study 

Dear Dr. Heikinheimo:

I'm pleased to inform you that your manuscript has been deemed suitable for publication in PLOS ONE. Congratulations! Your manuscript is now with our production department. 

Kind regards, 

on behalf of

Dr. Paola Viganò 

Academic Editor

PLOS ONE